# EFFICIENT CONVBN BLOCKS FOR TRANSFER LEARNING AND BEYOND

**Kaichao You**[†*]**, Guo Qin**[†]**, Anchang Bao**[†] **Meng Cao**[§]**, Ping Huang**[§]**, Jiulong Shan**[§]**,
Mingsheng Long**[†✉]
[†] School of Software, BNRist, Tsinghua University, China     [§] Apple
{ykc20,bac20,qing20}@mails.tsinghua.edu.cn
{mengcao,Huang_ping,jlshan}@apple.com
mingsheng@tsinghua.edu.cn

## ABSTRACT

Convolution-BatchNorm (ConvBN) blocks are integral components in various computer vision tasks and other domains. A ConvBN block can operate in three modes: Train, Eval, and Deploy. While the Train mode is indispensable for training models from scratch, the Eval mode is suitable for transfer learning and beyond, and the Deploy mode is designed for the deployment of models. This paper focuses on the trade-off between stability and efficiency in ConvBN blocks: Deploy mode is efficient but suffers from training instability; Eval mode is widely used in transfer learning but lacks efficiency. To solve the dilemma, we theoretically reveal the reason behind the diminished training stability observed in the Deploy mode. Subsequently, we propose a novel Tune mode to bridge the gap between Eval mode and Deploy mode. The proposed Tune mode is as stable as Eval mode for transfer learning, and its computational efficiency closely matches that of the Deploy mode. Through extensive experiments in object detection, classification, and adversarial example generation across 5 datasets and 12 model architectures, we demonstrate that the proposed Tune mode retains the performance while significantly reducing GPU memory footprint and training time, thereby contributing efficient ConvBN blocks for transfer learning and beyond. Our method has been integrated into both PyTorch (general machine learning framework) and MMCV/MMEngine (computer vision framework). Practitioners just need one line of code to enjoy our efficient ConvBN blocks thanks to PyTorch's builtin machine learning compilers.

## 1 INTRODUCTION

Feature normalization (Huang et al., 2023) is a critical component in deep convolutional neural networks to facilitate the training process by promoting stability, mitigating internal covariate shift, and enhancing network performance. BatchNorm (Ioffe & Szegedy, 2015) is a popular and widely adopted normalization module in computer vision. A convolutional layer (LeCun et al., 1998) together with a consecutive BatchNorm layer is often called a ConvBN block, which operates in three modes:

- Train mode. Mini-batch statistics (mean and standard deviation $\mu, \sigma$) are computed for feature normalization, and running statistics ($\hat{\mu}, \hat{\sigma}$) are tracked by exponential moving averages for testing individual examples when mini-batch statistics are unavailable.

- Eval mode. Running statistics are directly used for feature normalization without update, which is more efficient than Train mode, but requires tracked statistics to remain stable in training. It can also be used to validate models during development.

- Deploy mode. When the model does not require further training, computation in Eval mode can be accelerated (Markuš, 2018) by fusing convolution, normalization, and affine transformations into a single convolutional operator with transformed parameters. This is called Deploy mode, which produces the same output as Eval mode with better efficiency.

---

*: This work is conducted during Kaichao You's internship at Apple.
✉: Mingsheng Long is the corresponding author.

In Deploy mode, parameters for the convolution are computed once-for-all, removing batch normalization for faster inference during deployment.

The three modes of ConvBN blocks present a trade-off between computational efficiency and training stability, as shown in Table 1. Train mode is applicable for both train from scratch and transfer learning, while Deploy mode optimizes computational efficiency. Consequently, these modes traditionally align with three stages in deep models' lifecycle: Train mode for training, Eval mode for validation, and Deploy mode for deployment.

Table 1: Trade-off among modes of ConvBN blocks.

| Mode | Train | Eval | Tune (proposed) | Deploy |
|---|---|---|---|---|
| Train From Scratch | ✓ | ✗ | ✗ | ✗ |
| Transfer Learning | ✓ | ✓ | ✓ | ✗ |
| Training Efficiency | ★ | ★★ | ★★★ | ★★★ |

With the rise of transfer learning (Jiang et al., 2022), practitioners usually start with a pre-trained model, and instability of training from scratch is less of a concern. For instance, an object detector typically has one pre-trained backbone to extract features, and a head trained from scratch to predict bounding boxes and categories. Therefore, practitioners have started to explore Eval mode for transfer learning, which is more efficient than Train mode. Figure 1 presents the distribution of the normalization layers used in MMDetection (Chen et al., 2019), a popular object detection framework. In the context of transfer learning, a majority of detectors (496 out of 634) are trained with ConvBN blocks in Eval mode. Interestingly, our experiments suggest that Eval mode not only improves computational efficiency but also enhances the final performance over Train mode in certain transfer learning scenarios. For example, Appendix A shows training Faster-RCNN (Ren et al., 2015) on COCO (Lin et al., 2014) with Eval mode achieves significantly better mAP than Train mode, with either pre-trained ResNet101 backbone or pre-trained HRNet backbone.

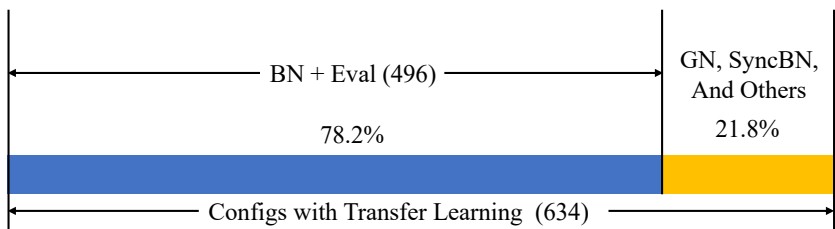

Figure 1: Usage of normalization layers in all the 634 object detectors with pre-trained backbones in the MMDetection framework (Chen et al., 2019). GN denotes GroupNorm, SyncBN represents synchronized BatchNorm across multiple GPUs and Eval indicates training ConvBN blocks in Eval mode. A majority of detectors (over 78%) are trained with ConvBN blocks in Eval mode.

Since transfer learning of ConvBN blocks in Eval mode has been a common practice, and forward calculation results between Deploy mode and Eval mode are equivalent, it is natural to ask if we can use Deploy mode for more efficient training. Unfortunately, Section 3.3 shows direct training in Deploy mode can lead to instability, as it is not designed for training.

In quest of efficient ConvBN blocks for transfer learning with pre-trained models, we theoretically uncover the underlying causes of training instability in Deploy mode and subsequently propose a novel Tune mode. It bridges the gap between Eval mode and Deploy mode, preserving functional equivalence with Eval mode in both forward and backward propagation while approaching the computational efficiency of Deploy mode. Our extensive experiments across transfer learning tasks (object detection and classification) and beyond (adversarial example generation) confirm the reduction in memory footprint and wall-clock training time without sacrificing performance.

Our contributions are summarized as follows:

- We theoretically analyze why Deploy mode is unstable for training, and propose efficient ConvBN blocks with a Tune mode to take advantages from both Eval and Deploy modes.

- We present extensive experiments across 12 models and 5 datasets to confirm the gain of Tune mode in transfer learning and beyond.
- Our method has been quickly integrated into open-source framework libraries like PyTorch and MMCV/MMEngine because of its evident benefit, improving the efficiency of hundreds of models for everyone using these frameworks.

## 2 RELATED WORK

### 2.1 NORMALIZATION LAYERS

Feature normalization has long been established in machine learning (Bishop, 2006), *e.g.*, Z-score normalization to standardize input features for smooth and isotropic optimization landscapes (Boyd & Vandenberghe, 2004). With the emergence of deep learning, normalization methods specifically tailored to intermediate activations, or feature maps, have been developed and gained traction.

Batch normalization (BN), proposed by Ioffe & Szegedy (2015), demonstrated that normalizing intermediate layer activations could expedite training and mitigate the effects of internal covariate shift. Since then, various normalization techniques have been proposed to address specific needs, such as group normalization (Wu & He, 2018) for small batch sizes, and layer normalization (Ba et al., 2016) typically employed in sequence models such as recurrent networks and Transformers. We direct interested readers to the survey by Huang et al. (2023) for an in-depth exploration of normalization layers.

Among various types of normalization, BatchNorm is a popular choice, partly due to its ability to be fused within convolution operations during deployment. Conversely, other normalization layers exhibit different behaviors compared to BatchNorm and often entail higher computational costs during deployment. The fusion allows ConvBN blocks to be efficiently deployed to massive edge and mobile devices where efficiency and power consumption control are critically important. This paper focuses on improving the efficiency of widely used ConvBN blocks for transfer learning and beyond.

### 2.2 VARIANTS OF BATCH NORMALIZATION

While batch normalization successfully improves training stability and convergence, it presents several limitations. These limitations originate from the different behavior during training and validation (Train mode and Eval mode), which is referred to as train-inference mismatch (Gupta et al., 2019) in the literature. Ioffe (2017) proposed batch renormalization to address the normalization issues with small batch sizes. Wang et al. (2019) introduced TransNorm to tackle the normalization problem when adapting a model to a new domain. Recently, researchers find that train-inference mismatch of BatchNorm plays an important role in test-time domain adaptation (Wang et al., 2021; 2022). In this paper, we focus on training ConvBN blocks in Eval mode, which is free of train-inference mismatch because its behavior is consistent in both training and inference.

Another challenge posed by BatchNorm is its memory intensiveness. While the computation of BatchNorm is relatively light compared with convolution, it occupies nearly the same memory as convolution because it records the feature map of convolutional output for back-propagation. To address this issue, Bulo et al. (2018) proposed to replace the activation function (ReLU (Nair & Hinton, 2010)) following BatchNorm with an invertible activation (such as Leaky ReLU (Maas et al., 2013)), thereby eliminating the need to store the output of convolution for backpropagation. However, this approach imposes an additional computational burden on backpropagation, as the input of activations must be recomputed by inverting the activation function. Their reduced memory footprint comes with the price of increased running time. In contrast, our proposed Tune mode effectively reduces both computation time and memory footprint for efficient transfer learning without any modification of network activations or any other architecture.

### 2.3 TRANSFER LEARNING

Training deep neural networks used to be difficult and time-consuming. Fortunately, with the advent of advanced network architectures like skip connections (He et al., 2016), and the availability of foundation models (Bommasani et al., 2022), practitioners can now start with pre-trained models and fine-tune them for various applications. Pre-trained models offer general representations (Donahue et al., 2014) that can accelerate the convergence of fine-tuning in downstream tasks. Consequently, the rule of thumb in computer vision tasks is to start with models pre-trained on large-scale datasets like ImageNet (Deng et al., 2009), Places (Zhou et al., 2018), or OpenImages (Kuznetsova et al.,

2018). This transfer learning paradigm alleviates the data collection burden required to build a deep model with satisfactory performance and can also expedite training, even if the downstream task has abundant data (Mahajan et al., 2018).

Train mode is the only mode for training ConvBN blocks from scratch. However, it is possible to use Eval mode for transfer learning, as we can exploit pre-trained statistics without updating them. Moreover, Eval mode is more computationally efficient than Train mode. Consequently, researchers usually fine-tune pre-trained models in Eval mode (Chen et al., 2019) (Figure 1), which maintains performance while offering improved computational efficiency. In this paper, we propose a novel Tune mode that further reduces memory footprint and training time while maintaining functional equivalence with the Eval mode during both forward and backward propagation. The Tune mode is a drop-in replacement of Eval mode while making ConvBN blocks more efficient.

### 2.4 MACHINE LEARNING COMPILERS

PyTorch (Paszke et al., 2019), a widely adopted deep learning framework, uses dynamic computation graphs that are built on-the-fly during computation. This imperative style of computation is user-friendly and leads to PyTorch's rapid rise in popularity. Nevertheless, the dynamic computation graphs complicate the speed optimization. Traditionally, operator analysis and fusion were only applied to models after training. The speed optimization usually involved a separate language or framework such as TensorRT (Vanholder, 2016) or other domain-specific languages, distinct from the Python language commonly employed for training. PyTorch has explored several ways, including symbolic tracing with `torch.fx` (Reed et al., 2022) and just-in-time tracing with `torch.jit`, to introduce machine learning compilers into the framework, and all the efforts are consolidated into PyTorch 2.0 (Wu, 2023). Leveraging PyTorch's pioneering compiler, our proposed Tune mode can automatically identify consecutive Convolution and BatchNorm layers without manual intervention.

## 3 METHOD

### 3.1 PROBLEM SETUP

In this paper, we study ConvBN blocks that are prevalent in various computer vision applications, especially in edge and mobile devices. A ConvBN block consists of two layers: (1) a convolutional layer with weight $\omega$ and bias $b$; (2) a BatchNorm layer with tracked mean $\hat{\mu}$ and standard deviation $\hat{\sigma}$, and weight $\gamma$ and bias $\beta$. We focus on the computation within ConvBN blocks, which is not affected by activation functions or skip connections after ConvBN blocks.

Given an input tensor $X$ with dimensions $[N, C_{\text{in}}, H_{\text{in}}, W_{\text{in}}]$, where $N$ represents the batch size, $C_{\text{in}}$ the number of input channels, and $H_{\text{in}}/W_{\text{in}}$ the spatial height/width of the input, a ConvBN block in Eval mode (the majority choice in transfer learning as shown in Figure 1) operates as follows. First, the convolutional layer computes an intermediate output tensor $Y = \omega \circledast X + b$ (we use $\circledast$ to denote convolution), resulting in dimensions $[N, C_{\text{out}}, H_{\text{out}}, W_{\text{out}}]$. Subsequently, the BN layer normalizes and applies an affine transformation to the intermediate output, producing the final output tensor $Z = \gamma \frac{Y - \hat{\mu}}{\sqrt{\hat{\sigma}^2 + \varepsilon}} + \beta$ with the same dimensions as $Y$.

Usually, the training loss consists of two parts: $J = J(Z)$ calculated on the network's output, and regularization loss $R$ calculated on the network's trainable parameters. Training is dominated by the gradient from $J(Z)$, especially at the beginning of training. The influence of $R$ is rather straightforward to analyze, since it directly and independently applies to each parameter. We omit the analysis for simplicity, as it does not change the main conclusion of this paper. Therefore, our primary focus lies in understanding the gradient with respect to the output loss function $J(Z)$ under different modes of ConvBN blocks. Note that $J(Z)$ can represent loss directly calculated on $Z$, as well as loss computed based on the output of subsequent layers operating on $Z$.

### 3.2 PRELIMINARY

#### 3.2.1 BACKWARD PROPAGATION OF CONVOLUTION

To discuss the stability of training, we must examine the details of backward propagation to understand the behavior of the gradient for each parameter. For a convolution layer with forward computation $Y = \omega \circledast X + b$, if the gradient back-propagated to $Y$ is $\frac{\partial J}{\partial Y}$, then the gradients of each input of the convolution layer, as explained in Bouvrie (2006), are: $\frac{\partial J}{\partial \omega} = \frac{\partial J}{\partial Y} \odot X$; $\frac{\partial J}{\partial X} = \omega_{\text{rot}} \circledast \frac{\partial J}{\partial Y}$; $\frac{\partial J}{\partial b} = \frac{\partial J}{\partial Y}$. The $\odot$ represents cross-correlation, and $\omega_{\text{rot}}$ is the rotated version of $\omega$, both are used to compute

the gradient of convolution (Rabiner & Gold, 1975). Note that these equations potentially contain broadcasting, a technique to allow element-wise arithmetic between two tensors with different shapes. Appendix B clarifies how broadcasting works in details.

### 3.2.2 ASSOCIATIVE LAW FOR CONVOLUTION AND AFFINE TRANSFORM

Convolution can essentially be viewed as a patch-wise matrix-vector multiplication, with the matrix (kernel weight) having a shape of $[C_{\text{out}}, k^2 C_{\text{in}}]$, and the vector having a shape of $[k^2 C_{\text{in}}]$. If an affine transform is applied to the weight along the $C_{\text{out}}$ dimension, then the affine transform is associative with the convolution operator. Formally speaking, $\gamma \cdot (\omega \circledast X) = (\gamma \cdot \omega) \circledast X$, where $\gamma$ is a $C_{\text{out}}$-dimensional vector multiplied to each row of the weight $\omega$. This association law lays the foundation of analyses for the Deploy mode and our proposed Tune mode. The associative law also applies to transposed convolution (Zeiler et al., 2010) and linear layers, therefore *the proposed Tune mode also works for TransposedConv-BN and Linear-BN blocks*.

With the necessary background established, we directly present the forward, backward, and memory footprint details in Table 2. Further analyses will be provided in subsequent sections.

Table 2: Computation graph of ConvBN blocks in different modes. Shape annotations for each tensor are available in Appendix C. We introduce Tune mode to improve the efficiency of ConvBN blocks, alleviating the dilemma between training stability and computational efficiency.

### 3.3 ANALYZING EVAL MODE AND DEPLOY MODE

With the help of equations in Table 2, the comparison between Eval mode and Deploy mode on efficiency and training stability is as straightforward as follows.

#### 3.3.1 FORWARD COMPUTATION EFFICIENCY

We first observe that Eval mode and Deploy mode have equivalent results in forward computation, and Deploy mode is more efficient. The equivalence can be proved by the definitions $\omega' = \frac{\gamma}{\sqrt{\hat{\sigma}^2+\epsilon}} \cdot \omega$ and $b' = (b - \hat{\mu}) \frac{\gamma}{\sqrt{\hat{\sigma}^2+\epsilon}} + \beta$, together with the associative law for convolution and affine transformations. However, Deploy mode pre-computes the weight $\omega'$ and $b'$, reducing the forward propagation to a single convolution calculation. Conversely, Eval mode requires a convolution, supplemented by a normalization and an affine transform on the convolutional output. This results in a slower forward propagation process for Eval mode. Moreover, Eval mode requires storing $X, Y$ for backward propagation, while Deploy mode only stores $X$. The memory footprint of Eval mode is nearly double of that in Deploy mode. Therefore, Deploy mode emerges as the more efficient of the two in terms of memory usage and computational time.

#### 3.3.2 TRAINING STABILITY

Our analyses suggest that Deploy mode tends to exhibit less training stability than Eval mode. Focusing on the convolutional weight, which constitutes the primary parameters in ConvBN blocks, we observe from Table 2 that the relationship of values and gradients between Deploy mode and Eval mode is $\omega' = \frac{\gamma}{\sqrt{\hat{\sigma}^2+\epsilon}}\omega$ and $\frac{\partial J}{\partial \omega'} = \frac{\sqrt{\hat{\sigma}^2+\epsilon}}{\gamma} \frac{\partial J}{\partial \omega}$. The scaling coefficients of the weight $(\frac{\gamma}{\sqrt{\hat{\sigma}^2+\epsilon}})$ are inverse of the scaling coefficients of the gradient $(\frac{\sqrt{\hat{\sigma}^2+\epsilon}}{\gamma})$. This can cause training instability in Deploy mode. For instance, if $\frac{\gamma}{\sqrt{\hat{\sigma}^2+\epsilon}}$ is small (say $0.1$), the weight reduces to one-tenth of its original value, while the gradient increases tenfold. This is a significant concern in real-world applications. As illustrated in Figure 2(a), these scaling coefficients range from as low as $0$ to as high as $30$, leading to unstable training. Figure 2(b) further substantiates this point through end-to-end experiments in both object detection and classification using Eval mode and Deploy mode. Training performance in Deploy mode is markedly inferior to that in Eval mode.

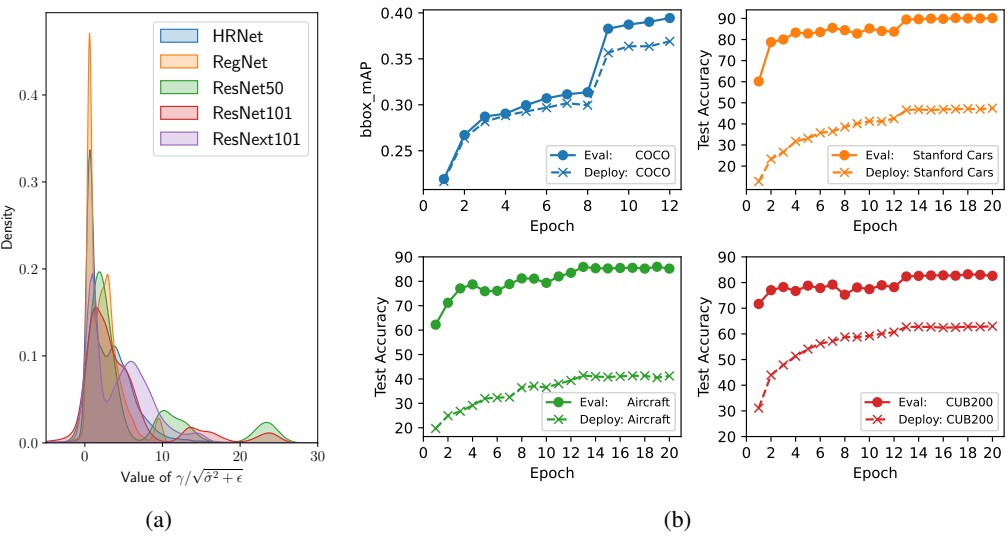

Figure 2: (a): Distribution of scaling coefficients for weight $\left(\gamma/\sqrt{\hat{\sigma}^2+\epsilon}\right)$ in different backbones. (b): Comparison between training with Eval mode and Deploy mode in both object detection and classification. Severe performance degradation is observed for training with Deploy mode.

In conclusion, *Deploy mode and Eval mode share the same forward calculation results, but present a dilemma in computation efficiency and training stability.*

### 3.4 TUNE MODE *v.s.* DEPLOY MODE AND EVAL MODE

Table 2 describes the detailed computation of the proposed Tune mode specifically designed for efficient transfer learning. This mode leverages the associative law of convolution and affine transformation, optimizing both memory and computation. *The main point is to calculate the transformed parameters dynamically, on-the-fly.* Next, we provide two critical analyses to show how the proposed Tune mode addresses the dilemma between training stability and computational efficiency, and how it bridges the gap between Eval mode and Deploy mode.

#### 3.4.1 TRAINING STABILITY

The associative law between convolution and affine transformation readily implies that the forward calculations between Eval mode and Tune mode are equivalent. The equivalence of backward calculations is less intuitive, particularly when considering the gradient of $\gamma$. To validate this, we employ an alternative approach: let $Z_1, Z_2$ represent the outputs of Eval mode and Tune mode, respectively. We define $Z_1 = Z_1(\omega, b, \gamma, \beta), Z_2 = Z_2(\omega, b, \gamma, \beta)$. Given that $Z_1 = Z_2$, and both are functions computed from the same set of parameters $(\omega, b, \gamma, \beta)$, we can assert that their Jacobian matrices are the same: $\frac{\partial Z_1}{\partial [\omega, b, \gamma, \beta]} = \frac{\partial Z_2}{\partial [\omega, b, \gamma, \beta]}$. This immediately suggests that both modes share the same backward propagation dynamics. Consequently, we can conclude that *Tune mode is as stable as Eval mode in transfer learning*.

#### 3.4.2 EFFICIENCY

According to Table 2, Eval mode requires saving the input feature map $X$ and the convolutional output $Y$, with total memory footprint $X + Y$ for each ConvBN block. In contrast, Tune mode stores $X$ and the transformed weights $\omega'$, with total memory footprint $X + \omega'$ for each ConvBN block. Since feature maps $Y$ are usually larger than convolutional weights $\omega'$, this difference signifies that Tune mode requires less memory for training. The same applies to the analysis of computation: computation in Eval mode consists of a convolution followed by an affine transformation on the *convolutional feature map $Y$*; Tune mode computation consists of an affine transformation on the original *convolutional weights $\omega$* succeeded by a convolution with the transformed weights $\omega'$. An affine transformation on convolutional weights executes faster than on feature maps. Therefore, Tune mode outperforms Eval mode both in memory usage and computation speed. Please refer to Appendix M for formal analyses of efficiency using the $\mathcal{O}$ notation.

The above conclusion can be empirically validated using a standard ResNet-50 (He et al., 2016) model with variable batch sizes and input sizes. The results, displayed in Figure 3, clearly indicate that *Tune mode is more efficient than Eval mode* across all tested settings. The memory footprint of Tune mode consumed by pre-trained backbone in transfer learning can be reduced to one half of that in Eval mode, and the computation time is reduced by about $10\%$. The comparison between Tune and Deploy in efficiency can be found in Appendix D, they are nearly the same in terms of efficiency, but Deploy mode is less stable and incurs worse accuracy than Tune mode.

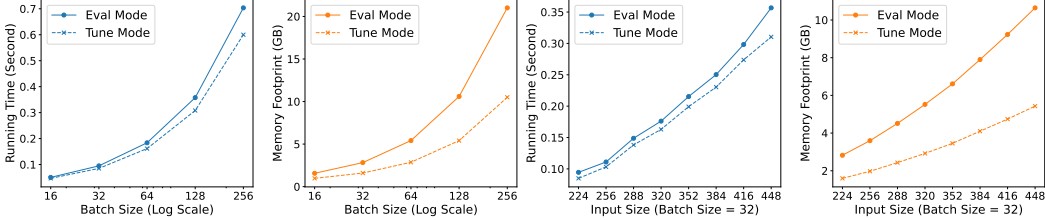

Figure 3: Memory footprint and running time comparison for Eval mode and Tune mode. The base setting is batchsize $= 32$ and input dimension $= 224 \times 224$, and we vary batchsize and input dimension to test the efficiency.

The comparison among Eval/Tune/Deploy can be summarized as follows:

- Deploy mode $\approx$ Tune mode $>$ Eval mode, in terms of efficiency.
- Eval mode $=$ Tune mode $>$ Deploy mode, in terms of training stability.

Therefore, the proposed Tune mode successfully bridges the gap between Eval mode and Tune mode, improving the efficiency of ConvBN blocks with Eval mode while keeping the stability of training.

## 4 EXPERIMENTS

Our algorithm has been tested against 5 datasets and 12 model architectures, as summarized in Appendix E. The total computation for results reported in this paper is about 3400 hours of V100 GPU (32GB) counted by our internal computing infrastructure. More details are given in Appendix F.

Extensive experiments confirm the benefit of our method, and convince the community of PyTorch and MMCV/MMEngine (Contributors, 2018; 2022) to quickly integrate our method. Consequently, anyone using either library can enjoy the benefit of our algorithm with as simple as a one-line code change. We provide guidelines for each library to turn on the Tune mode in Appendix N.

Since Tune mode improves efficiency over Eval mode, our experiments focus on the efficiency and accuracy comparison between Eval mode and Tune mode. Train mode is often inferior in efficiency and Deploy mode is often inferior in accuracy, so we don't include them in the main paper but leave the full comparison among four modes in Appendix G and H.

### 4.1 OBJECT CLASSIFICATION IN TRANSFER LEARNING

We first show the benefit of Tune mode in object classification in transfer learning. We use a popular open-source TLlib (Jiang et al., 2022), and the datasets include CUB-200 (Wah et al., 2011) for fine-grained bird classification, Standford Cars (Krause et al., 2013) and Aircrafts (Maji et al., 2013). The network backbone is ResNet50 pre-trained on ImageNet. Each experiment is repeated three times with different random seeds to report mean and standard deviation. Results are reported in Table 3, with further details in Appendix G. Compared with Eval mode, the proposed Tune mode *reduces more than* $7\%$ *computation time and* $36\%$ *memory footprint*.

Table 3: Results for Tune mode in classification using TLlib.

| Dataset | mode | Accuracy | Memory (GB) | Time (second/iteration) |
|---|---|---|---|---|
| CUB-200 | Eval | 82.62 ($\pm$ 0.14) | 19.499 | 0.549 |
| | Tune | 83.20 ($\pm$ 0.00) | 12.323 (**36.80%**↓) | 0.501 (**8.74%**↓) |
| Aircrafts | Eval | 85.21 ($\pm$ 0.22) | 19.497 | 0.548 |
| | Tune | 85.90 ($\pm$ 0.26) | 12.321 (**36.81%**↓) | 0.505 (**7.85%**↓) |
| Stanford Cars | Eval | 90.11 ($\pm$ 0.03) | 19.499 | 0.541 |
| | Tune | 90.13 ($\pm$ 0.12) | 12.321 (**36.81%**↓) | 0.491 (**9.24%**↓) |

Table 4: Object Detection results on different detectors and backbones.

| Detector | Backbone | BatchSize | Precision | mode | mAP | Memory (GB) |
|---|---|---|---|---|---|---|
| Faster RCNN | ResNet50 | 2 | FP32 | Eval | 0.3739 | 3.857 |
| | | | | Tune | 0.3728 (-0.0011) | 3.003 (**22.15%**↓) |
| Mask RCNN | ResNet50 | 2 | FP32 | Eval | 0.3824 | 4.329 |
| | | | | Tune | 0.3825 (+0.0001) | 3.470 (**19.85%**↓) |
| Mask RCNN | ResNet101 | 16 | FP16 | Eval | 0.3755 | 13.687 |
| | | | | Tune | 0.3756 (+0.0001) | 9.980 (**27.08%**↓) |
| Retina Net | ResNet50 | 2 | FP32 | Eval | 0.3675 | 3.631 |
| | | | | Tune | 0.3647 (-0.0028) | 2.774 (**23.59%**↓) |
| Faster RCNN | ResNet101 | 2 | FP32 | Eval | 0.3944 | 5.781 |
| | | | | Tune | 0.3921 (-0.0023) | 4.183 (**27.65%**↓) |
| Faster RCNN | ResNext101 | 2 | FP32 | Eval | 0.4126 | 6.980 |
| | | | | Tune | 0.4131 (+0.0005) | 4.773 (**31.62%**↓) |
| Faster RCNN | RegNet | 2 | FP32 | Eval | 0.3985 | 4.361 |
| | | | | Tune | 0.3995 (+0.0010) | 3.138 (**28.06%**↓) |
| Faster RCNN | HRNet | 2 | FP32 | Eval | 0.4017 | 8.504 |
| | | | | Tune | 0.4031 (+0.0014) | 5.463 (**35.76%**↓) |
| Faster RCNN | RepVGG | 16 | FP16 | Eval | 0.3350 | 15.794 |
| | | | | Tune | 0.3350 (+0.0000) | 8.996 (**43.04%**↓) |

## 4.2 Object Detection in Transfer Learning

This section presents object detection results on the widely used COCO (Lin et al., 2014) dataset. The MMDetection library uses Eval mode by default, and we compare the results by switching models to Tune mode. We test against various mainstream CNN backbones and detection algorithms (including Faster RCNN (Ren et al., 2015), Mask RCNN (He et al., 2017), and Retina Net (Lin et al., 2017)). Results are displayed in Table 4, with additional results available in Appendix H. Object detection experiments are costly, and therefore we do not repeat three times to calculate mean and standard deviation. Appendix I shows that the standard deviation of performance across different runs is as small as 0.0005. The change of mAP in Table 4 falls into the range of random fluctuation across experiments.

With different architecture, batch size and training precision (Micikevicius et al., 2018), *Tune mode has almost the same mAP as Eval mode, while remarkably reducing the memory footprint by about* $20\% \sim 40\%$. Note that detection models typically have a pre-trained backbone for extracting features, and a head trained from scratch for producing bounding boxes and classification. The head consumes the major computation time, and the backbone consumes the major memory footprint. Because ConvBN blocks mainly lie in the backbone, our Tune mode mainly benefits the backbone, therefore reducing only the memory footprint. Computation speedup is not obvious in objection detection, and we only report the reduction of memory footprint here.

## 4.3 Application of Tune Mode Beyond Transfer Learning

Our method is designed for transfer learning. However, we find that its application can go beyond transfer learning. Any model using Eval mode can benefit from our Tune mode. Adversarial example generation (Szegedy et al., 2013) is a representative application of our method: when generating adversarial examples for adversarial training (Goodfellow et al., 2015), an important step is to calculate the gradient $\nabla_x \mathcal{L}(\theta, x, y)$ with respect to inputs $x$, given inputs, labels $y$, and parameters $\theta$. Common techniques for producing adversarial samples, such as FGSM (Goodfellow et al., 2015), BIM (Kurakin et al., 2016), and PGD (Madry et al., 2017), all perturb the inputs based on the gradients, where the model is in Eval mode. Turning on Tune mode can improve the efficiency of adversarial sample generation. Concretely, we perform forward and backward propagation of samples through the model to compute the gradient of input, and measure the time cost as well as GPU memory footprint. The experimental results can be found in Figure 4, with detailed numbers available in Appendix J. Across different models, Tune mode can *achieve 5%-8% speedup and save 30%-45% of GPU memory*. These experiments cover widely used network architectures, and also cover UNet that has transposed convolution layers, demonstrating the broad application of our method.

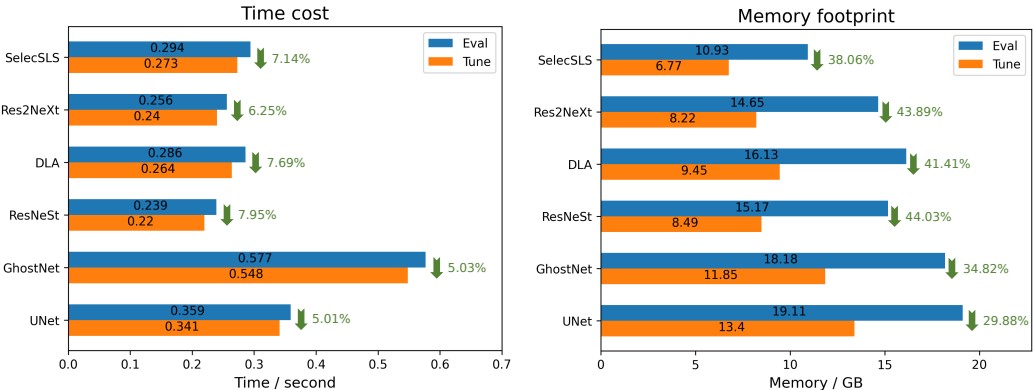

Figure 4: Tune mode *v.s.* Eval mode in adversarial example generation.

## 5 Conclusion

This paper proposes efficient ConvBN blocks with a novel Tune mode for transfer learning and beyond. Tune Mode is equivalent to Eval mode in both forward and backward calculation while reducing memory footprint and computation time without hurting performance. Our experiments confirm the benefit across dozens of models and tasks, reducing at most 44% memory footprint and 9% computation time. We further bring the proposed method into open-source frameworks the community uses everyday, reducing the cost of training networks with ConvBN blocks.

ACKNOWLEDGMENTS

We would like to thank many open-source contributors for helping the adoption of this technique into PyTorch, MMDetection, and MMCV, including Jason Ansel from Meta, and Wenwei Zhang, Haochen Ye, Zaida Zhou from OpenMMLab.

This work was supported by the National Key Research and Development Plan (2021YFB1715200), the National Natural Science Foundation of China (U2342217 and 62022050), the BNRist Innovation Fund (BNR2024RC01010), and the National Engineering Research Center for Big Data Software.

Kaichao You is partly supported by the Apple Scholar in AI/ML.

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

## A    COMPARISON OF TRAIN AND EVAL FOR OBJECT DETECTION

We compared the performance of detection models trained in Train Mode and Eval Mode, using two backbones (Resnet101 and HRNet). Results are shown in Table 5 and the training curves are shown in Figure 5.

To ensure fair comparison, we use official training schemes from MMDetection: faster_rcnn/faster_rcnn_r50_fpn_1x_coco.py and hrnet/faster_rcnn_hrnetv2p_w32_1x_coco.py. The default choice in MMDetection is training with Eval Mode, and we only change `model.backbone.norm_eval=False` to switch training to Train Mode.

These results indicate that *Eval Mode sometimes outperforms Train Mode in transfer learning of object detection.*

Table 5: mAP of Faster RCNN trained under different ConvBN block Mode.

| Configuration File | Eval Mode | Train Mode |
|---|---|---|
| faster_rcnn/faster_rcnn_r50_fpn_1x_coco.py | 0.3944 | 0.3708 |
| hrnet/faster_rcnn_hrnetv2p_w32_1x_coco.py | 0.4017 | 0.3828 |

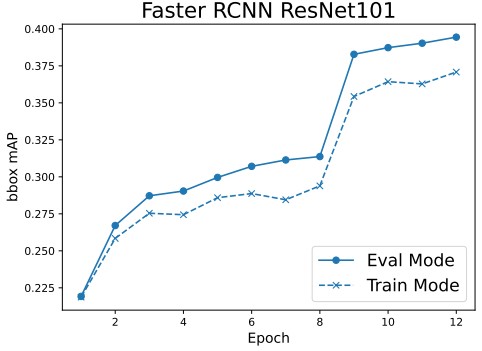 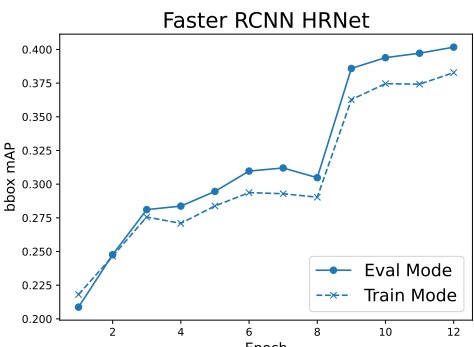

Figure 5: Training curve of Faster RCNN with *ResNet101* and *HRNet* backbone. Models trained in Train Mode shows noticeable performance deterioration compared to Eval Mode.

## B    BACKWARD PROPAGATION OF BROADCAST

Take the convolution operation as an example: the convolutional output $Y$ has a shape of $[N, C_{\text{out}}, H_{\text{out}}, W_{\text{out}}]$, while the tracked mean $\hat{\mu}$ has a shape of $[C_{\text{out}}]$, and $Y - \hat{\mu}$ implies first replicating $\hat{\mu}$ to have a shape of $[N, C_{\text{out}}, H_{\text{out}}, W_{\text{out}}]$, then performing element-wise subtraction. This can be explained by introducing an additional broadcast operator $\mathcal{B}_Y$, where $\mathcal{B}_Y(\hat{\mu})$ broadcasts $\hat{\mu}$ to match the shape of $Y$. The underlying calculation is actually $Y - \mathcal{B}_Y(\hat{\mu})$. The backward calculation for the broadcast operator $\frac{\partial \mathcal{B}_Y(\hat{\mu})}{\partial \hat{\mu}}$ is the reverse of replication, i.e., summing over a large tensor with the shape of $[N, C_{\text{out}}, H_{\text{out}}, W_{\text{out}}]$ into a smaller tensor with the shape of $[C_{\text{out}}]$. This helps understand the backward equation for the bias term $\frac{\partial J}{\partial b} = \frac{\partial J}{\partial Y}$, which actually means $\frac{\partial J}{\partial b} = \frac{\partial \mathcal{B}_Y(b)}{\partial b} \frac{\partial J}{\partial Y}$, *i.e.*, summing $\frac{\partial J}{\partial Y}$ to be compatible with the shape of $\frac{\partial J}{\partial b}$. Due to the prevalence of broadcast in neural networks, we omit them to simplify equations.

## C  CODE DETAILS OF TRAIN/EVAL/DEPLOY MODE

Cmputation details of ConvBN blocks in different modes, with shape annotations for each tensor available in the following code snippet.

```
# input: A faeture map X. The shape of X is [N, C_in, H_in, W_in].
# input: A convolutional layer "conv" with kernel-size k and output channel number C_out; it
    has a weight parameter W with the shape of [C_out, C_in, k, k] and a bias parameter b with
    the shape of [C_out].
# input: A BatchNorm layer "bn" with output channel number C_out; it has a weight parameter γ
    with the shape of [C_out], and a weight parameter β with the shape of [C_out]. The momentum
    update rate of BatchNorm is a constant number α ∈ (0, 1).
# output: Z = bn(conv(X))

# code explained in a pytorch style
import torch

"Train Mode"
# calculate the output of convolution
Y = W ⊛ X + b  # ⊛ for convolution. The shape of Y is [N, C_out, H_out, W_out]
# calculate the mean for normalization
μ = torch.mean(Y, dim=(0, 2, 3)) # μ has a shape of [C_out]
# calculate the variation for normalization
σ² = torch.var(Y, dim=(0, 2, 3)) # σ² has a shape of [C_out]
# update tracked statistics, μ̂ and σ̂² keep track of moving mean and moving variance. They are
    initialized to 0 and 1 respectively if the model is trained from scratch, or are
    inherited from pre-trained values.
μ̂ ← μ̂ + α(μ − μ̂)
σ̂² ← σ̂² + α(σ² − σ̂²)

# normalize the output
# ε is a small positive number to avoid zero division
Ȳ = (Y−μ)/√(σ²+ε)  # μ and σ² are broadcast to match the shape of Y
# apply the affine transform
Z = γ * Ȳ + β  # γ and β are broadcast to match the shape of Ȳ

"Eval Mode"
# calculate the output of convolution
Y = W ⊛ X + b
# normalize the output with tracked statistics
Ȳ = (Y−μ̂)/√(σ̂²+ε)
# apply the affine transform
Z = γ * Ȳ + β

"Deploy Mode"
# update the weight and bias of the convolution once for all
Ŵ = W * γ/√(σ̂²+ε)
b̂ = (b − μ̂) γ/√(σ̂²+ε) + β

# convolution with updated parameters is equivalent to consecutive convolution and batch
    normalization
Z = Ŵ ⊛ X + b̂
```

Listing 1: Computation details for consecutive Convolution and BatchNorm layers in different modes

## D EFFICIENCY COMPARISON BETWEEN TUNE AND DEPLOY

Table 6: Efficiency comparison between Eval, Tune and Deploy. Time is measured by second/iteration and the memory is the peak memory footprint (GB) during forward and backward. We can observe that Deploy ≈ Tune > Eval in terms of efficiency.

| Batch Size | Input Size | Eval Mode | | Tune Mode | | DeployMode | |
|---|---|---|---|---|---|---|---|
| | | Time | Memory | Time | Memory | Time | Memory |
| 32 | 224 | 0.0945 | 2.8237 | 0.0849 | 1.5973 | 0.0830 | 1.5619 |
| 32 | 256 | 0.1110 | 3.5965 | 0.1032 | 1.9732 | 0.1011 | 1.9416 |
| 32 | 288 | 0.1488 | 4.5130 | 0.1382 | 2.4325 | 0.1356 | 2.3728 |
| 32 | 320 | 0.1761 | 5.5216 | 0.1630 | 2.9207 | 0.1609 | 2.8363 |
| 32 | 352 | 0.2153 | 6.6120 | 0.1991 | 3.4546 | 0.1969 | 3.3682 |
| 32 | 384 | 0.2503 | 7.9005 | 0.2304 | 4.0995 | 0.2281 | 4.0219 |
| 32 | 416 | 0.2983 | 9.2317 | 0.2738 | 4.7429 | 0.2721 | 4.6640 |
| 32 | 448 | 0.3567 | 10.6448 | 0.3104 | 5.4306 | 0.3077 | 5.3421 |
| 16 | 224 | 0.0505 | 1.5671 | 0.0467 | 0.9727 | 0.0448 | 0.9397 |
| 32 | 224 | 0.0948 | 2.8237 | 0.0849 | 1.5973 | 0.0831 | 1.5631 |
| 64 | 224 | 0.1837 | 5.4125 | 0.1613 | 2.8617 | 0.1590 | 2.7808 |
| 128 | 224 | 0.3577 | 10.6001 | 0.3081 | 5.4088 | 0.3060 | 5.3284 |
| 256 | 224 | 0.7035 | 21.0107 | 0.6001 | 10.5011 | 0.5966 | 10.4234 |

## E MODELS AND DATASETS TESTED

We conduct extensive experiments in object detection, classification, and adversarial example generation.

Our experiments cover 5 datasets:

- CUB-200 (Wah et al., 2011): CUB-200 is the most widely-used dataset for fine-grained visual categorization task. It contains 11,788 images of 200 subcategories belonging to birds.

- Standford Cars (Krause et al., 2013): Standford Cars consists of 196 classes of cars with a total of 16,185 images.

- Aircrafts (Maji et al., 2013): Aircrafts contains 10,200 images of aircraft, with 100 images for each of 102 different aircraft model variants.

- COCO (Lin et al., 2014): COCO is a large-scale object detection, segmentation, key-point detection, and captioning dataset released by Microsoft. The dataset consists of 328K images.

- ImageNet (Deng et al., 2009): ImageNet dataset contains 14,197,122 annotated images according to the WordNet hierarchy and it is instrumental in advancing computer vision and deep learning research.

Our experiments cover 12 model architectures:

- ResNet50, ResNet101 (He et al., 2016): ResNet introduced the residual structure, making it possible to train models with hundreds or thousands of layers, which was a significant breakthrough in deep learning.

- ResNeXt101 (Xie et al., 2017): ResNeXt is constructed by repeating a building block that aggregates a set of transformations with the same topology and achieved second place in ILSVRC 2016.

- RegNet (Radosavovic et al., 2020): RegNet is a self-regulated network for image classification and can be easily implemented and appended to any ResNet architecture.

- HRNet (Sun et al., 2019): HRNet is a general purpose convolutional neural network for tasks like semantic segmentation, object detection and image classification and is able to maintain high resolution representations through the whole process.

- RepVGG (Ding et al., 2021): RepVGG is a simple but powerful architecture of convolutional neural network, which has a VGG-like inference-time body composed of nothing but a stack of 3x3 convolution and ReLU.

- SelecSLS (Mehta et al., 2020): SelecSLS uses novel selective long and short range skip connections to improve the information flow allowing for a drastically faster network without compromising accuracy.

- Res2NeXt (Gao et al., 2019): Res2NeXt represents multi-scale features at a granular level and increases the range of receptive fields for each network layer.

- DLA (Yu et al., 2018): Extending "shallow" skip connections, DLA incorporates more depth and sharing. It contains iterative deep aggregation (IDA) and hierarchical deep aggregation (HDA).

- ResNeSt (Zhang et al., 2022): ResNeSt applies the channel-wise attention on different network branches to leverage their success in capturing cross-feature interactions and learning diverse representations.

- GhostNet (Han et al., 2020): GhostNet is a type of convolutional neural network that is built using Ghost modules, which aim to generate more features by using fewer parameters.

- UNet (Ronneberger et al., 2015): UNet consists of a contracting path and an expansive path and is widely employed across various facets of semantic segmentation.

## F  ESTIMATION OF TOTAL COMPUTATION USED IN THIS PAPER

Each trial of classification experiments in Section 4.1 requires about 2 hours of V100 GPU training. The numbers reported in this paper requires 18 trials, which cost about 36 GPU hours.

Each trial of detection experiments in Section 4.2 requires about 12 hours of 8 V100 GPU training, which is 96 GPU hours. The numbers reported in this paper requires 25 trials, which cost about 2400 GPU hours.

Each trial of pre-training experiments in Section K requires about 24 hours of 8 V100 GPU training, which is 192 GPU hours. The numbers reported in this paper requires about $\frac{5}{3}$ full trials, which cost about 320 GPU hours.

Some experiments for the purpose of analyses also cost computation. Figure 2 requires two trials of object detection and 6 trials of object classification, with about 204 GPU hours. Figure 5 requires four trials of object detection, with about 384 GPU hours.

Summing the above numbers up, and considering all the fractional computation for the rest analyses experiments, *this paper costs about* 3400 *GPU hours*.

Considering the cost of prototyping and previous experiments that do not get into the paper, *the total cost of this project is about* 5000 *GPU hours*.

Note that these numbers are rough estimation of the cost, and do not include the additional cost for storing data/system maintenance etc.

## G  COMPARISON OF FOUR MODES FOR OBJECT CLASSIFICATION

The below settings are taken from the default values in the TLlib library: ResNet50 is the backbone network and all parameters are optimized by Stochastic Gradient Descent with 0.9 momentum and 0.0005 weight decay. Each training process consisted of 20 epochs, with 500 iterations per epoch. We set the initial learning rates to 0.001 and 0.01 for the feature extractor and linear projection head respectively, and scheduled the learning rates of all layers to decay by 0.1 at epochs 8 and 12. The input images were all resized and cropped to $448 \times 448$, and the batch size was fixed at 48. Since the backbone network takes the major computation, the memory and time in three different dataset are very similar.

Table 7: Comparison of four modes in classification using TLlib.

| Dataset | mode | Accuracy | Memory (GB) | Time (second/iteration) |
|---|---|---|---|---|
| CUB-200 | Train | 83.07 | 19.967 | 0.571 |
| | Eval | 82.62 | 19.499 | 0.549 |
| | Deploy | 62.96 | 12.002 | 0.511 |
| | Tune | 83.20 | 12.323 | 0.501 |
| Aircrafts | Train | 85.40 | 19.965 | 0.564 |
| | Eval | 85.21 | 19.497 | 0.548 |
| | Deploy | 41.22 | 12.000 | 0.506 |
| | Tune | 85.90 | 12.321 | 0.505 |
| Stanford Cars | Train | 89.87 | 19.967 | 0.571 |
| | Eval | 90.11 | 19.499 | 0.541 |
| | Deploy | 47.42 | 12.002 | 0.507 |
| | Tune | 90.13 | 12.321 | 0.491 |

## H   DETAILED OBJECT DETECTION EXPERIMENTAL RESULTS

For object detection, more detailed comparison of Eval mode and Tune mode is presented in Table 8, while a comparison of the four modes can be found in Table 9.

Table 8: Detailed Object Detection experimental results.

| Detector | Backbone | BatchSize | Precision | mode | mAP | Memory(GB) |
|---|---|---|---|---|---|---|
| Faster RCNN | ResNet50 | 2 | FP32 | Eval | 0.3739 | 3.857 |
| | | | | Tune | 0.3728 (-0.0011) | 3.003 (**22.15%↓**) |
| Mask RCNN | ResNet50 | 2 | FP32 | Eval | 0.3824 | 4.329 |
| | | | | Tune | 0.3825 (+0.0001) | 3.470 (**19.85%↓**) |
| Mask RCNN | ResNet101 | 16 | FP16 | Eval | 0.3755 | 13.687 |
| | | | | Tune | 0.3756 (+0.0001) | 9.980 (**27.08%↓**) |
| Retina Net | ResNet50 | 2 | FP32 | Eval | 0.3675 | 3.631 |
| | | | | Tune | 0.3647 (-0.0028) | 2.774 (**23.59%↓**) |
| Faster RCNN | ResNet101 | 2 | FP32 | Eval | 0.3944 | 5.781 |
| | | | | Tune | 0.3921 (-0.0023) | 4.183 (**27.65%↓**) |
| Faster RCNN | ResNet101 | 2 | FP16 | Eval | 0.3944 | 3.849 |
| | | | | Tune | 0.3925 (-0.0019) | 3.138 (**18.47%↓**) |
| Faster RCNN | ResNet101 | 8 | FP16 | Eval | 0.3922 | 10.411 |
| | | | | Tune | 0.3917 (-0.0005) | 7.036 (**32.41%↓**) |
| Faster RCNN | ResNet101 | 16 | FP16 | Eval | 0.3902 | 19.799 |
| | | | | Tune | 0.3899 (-0.0003) | 12.901(**34.83%↓**) |
| Faster RCNN | ResNext101 | 2 | FP32 | Eval | 0.4126 | 6.980 |
| | | | | Tune | 0.4131 (+0.0005) | 4.773 (**31.62%↓**) |
| Faster RCNN | RegNet | 2 | FP32 | Eval | 0.3985 | 4.361 |
| | | | | Tune | 0.3995 (+0.0010) | 3.138 (**28.06%↓**) |
| Faster RCNN | HRNet | 2 | FP32 | Eval | 0.4017 | 8.504 |
| | | | | Tune | 0.4031 (+0.0014) | 5.463 (**35.76%↓**) |
| Faster RCNN | RepVGG | 16 | FP16 | Eval | 0.3350 | 15.80 |
| | | | | Tune | 0.3350 (+0.0000) | 9.00 (**43.04%↓**) |

Table 9: Comparison of four modes in detection.

| Detector | Backbone | Batchsize | Precision | mode | mAP | Memory(GB) | Time(sec/iter) |
|---|---|---|---|---|---|---|---|
| Faster RCNN | ResNet101 | 2 | FP32 | Train | 0.3708 | 5.782 | 0.3116 |
| | | | | Eval | 0.3944 | 5.781 | 0.3060 |
| | | | | Deploy | 0.3690 | 4.02 | 0.3060 |
| | | | | Tune | 0.3921 | 4.18 | 0.3085 |

## I  SMALL RANDOMNESS IN OBJECT DETECTION

The high computational cost limited us to repeating the experiment only once for validating small randomness in object detection. We conducted three trials of the Faster RCNN ResNet50 standard configuration and obtained an average best mAP of 0.3748, 0.3735, and 0.3739, with a standard deviation of 0.000543. These results demonstrate that object detection tasks have very little randomness.

## J  ADVERSARIAL EXAMPLE GENERATION

Table 10 presents more detailed model information and batch size information for the adversarial example generation experiments.

Table 10: Adversarial example generation result.

| Arch | | ResNeSt101e | DLA102 | SelecSLS42b | Res2NeXt50 | GhostNet_100 | UNet |
|---|---|---|---|---|---|---|---|
| Batch size | | 64 | 128 | 256 | 128 | 512 | 128 |
| Time (s) | Eval | 0.239 | 0.286 | 0.294 | 0.256 | 0.577 | 0.359 |
| | Tune | 0.22 (**7.95%**↓) | 0.264(**7.69%**↓) | 0.273 (**7.14%**↓) | 0.240 (**6.25%**↓) | 0.548 (**5.03%**↓) | 0.341(**5.01%**↓) |
| Memory (GB) | Eval | 15.17 | 16.13 | 10.93 | 14.65 | 18.18 | 19.11 |
| | Tune | 8.49 (**44.03%**↓) | 9.45 (**41.41%**↓) | 6.77 (**38.06%**↓) | 8.22 (**43.89%**↓) | 11.85 (**34.82%**↓) | 13.40 (**29.88%**↓) |

## K  TUNE MODE CONVBN FOR PRE-TRAINING

Tune mode is designed for transfer learning because it requires tracked statistics to normalize features. Here we show that Tune mode can also be used in late stages of pre-training. We use the prevalent ImageNet pre-training as baseline, which has three stages with decaying learning rate. We tried to turn on Tune mode at the third stage, the accuracy slightly dropped. Nevertheless, due to our implementation with `torch.fx`, we can dynamically switch the mode during training. Therefore, we also tried to alternate between Train mode and Tune mode at the third stage, which retained the accuracy with less computation time.

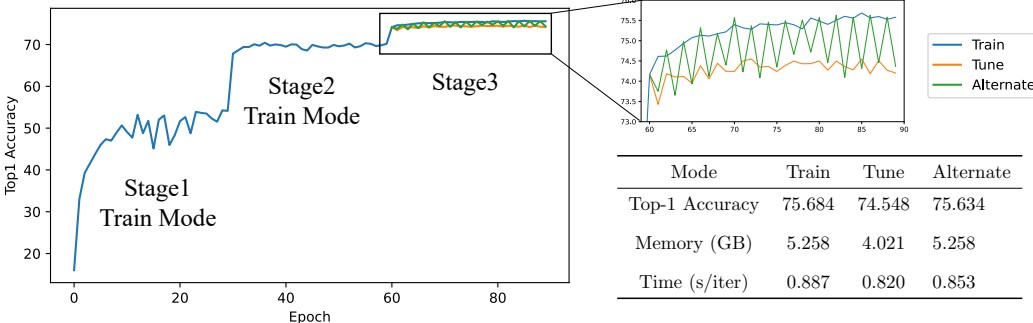

| Mode | Train | Tune | Alternate |
|---|---|---|---|
| Top-1 Accuracy | 75.684 | 74.548 | 75.634 |
| Memory (GB) | 5.258 | 4.021 | 5.258 |
| Time (s/iter) | 0.887 | 0.820 | 0.853 |

Figure 6: ImageNet Pre-training Results

## L  COMPARING WITH ALTERNATIVES TO REDUCING MEMORY FOOTPRINT

### L.1  FROZENBATCHNORM

FrozenBatchNorm, as used in Detectron2 (Wu et al., 2019), freezes the weight and bias of Batch-Norm layers, reducing memory footprint at the cost of less trainable parameters, which limits the

network's expressive power and hinders the accuracy of trained models. As shown in Table 11, FrozenBatchNorm's mAP in detection is lower than baseline, while our Tune mode doesn't hurt mAP.

Table 11: Comparision between proposed detection baseline and FrozenBatchNorm. FrozenBatch-Norm incurs performance loss.

|  | Faster RCNN ResNet 50 2X | Mask RCNN ResNet 50 2X |
|---|---|---|
| Baseline | 0.3832 | 0.3911 |
| FrozenBatchNorm | 0.3790 | 0.3904 |

## L.2 INPLACE-ABN

We compare our proposed Tune Mode with the Inplace-ABN (Bulo et al., 2018), a memory reduction method for ConvBN blocks by invertible activations. We show transfer learning experiments on TLlib (Jiang et al., 2022) with the same settings of section 4.1.

To apply Inplace-ABN blocks, we find BN-ReLU patterns in the pretrained network using `torch.fx` and replace them with the Inplace-ABN blocks provided by Bulo et al. (2018). Results are summarized in Table 12. Inplace-ABN block saves approximately 16% memory at the cost of 8% additional computation time, and also hurts the accuracy significantly as it modifies the network architecture. Compared to Inplace-ABN, our proposed Tune mode saves more memory footprint and requires less computation time, while retaining the accuracy.

Table 12: Comparision between proposed Tune Mode and the Inplace-ABN (Bulo et al., 2018). We can observe that Inplace-ABN saves memory at the cost of speed and accuracy, while Tune mode saves memory at no cost of computation time or accuracy.

| Dataset | Method | Accuracy | Memory (GB) | Time (second/iteration) |
|---|---|---|---|---|
| CUB-200 | Baseline | $83.07_{\pm0.15}$ | 19.967 | 0.571 |
|  | Inplace-ABN | $74.80_{\pm0.21}$ (8.27$\downarrow$) | 16.739 (16.17%$\downarrow$) | 0.623 (9.10%$\uparrow$) |
|  | Tune Mode (ours) | $\mathbf{83.20}_{\pm0.00}$ (0.13$\uparrow$) | 12.323 (**38.28%**$\downarrow$) | 0.501 (**12.26%**$\downarrow$) |
| Aircraft | Baseline | $85.40_{\pm0.20}$ | 19.965 | 0.564 |
|  | Inplace-ABN | $78.23_{\pm0.45}$ (7.17$\downarrow$) | 16.737 (16.17%$\downarrow$) | 0.620 (8.58%$\uparrow$) |
|  | Tune Mode (ours) | $\mathbf{85.90}_{\pm0.26}$ (0.50$\uparrow$) | 12.323 (**38.28%**$\downarrow$) | 0.505 (**10.51%**$\downarrow$) |
| Stanford Car | Baseline | $89.87_{\pm0.06}$ | 19.967 | 0.571 |
|  | Inplace-ABN | $86.30_{\pm0.28}$ (3.57$\downarrow$) | 16.739 (16.17%$\downarrow$) | 0.614 (7.53%$\uparrow$) |
|  | Tune Mode (ours) | $\mathbf{90.13}_{\pm0.12}$ (0.26$\uparrow$) | 12.321 (**38.28%**$\downarrow$) | 0.491 (**14.00%**$\downarrow$) |

## M   THEORETICAL ANALYSES OF BENEFIT IN MEMORY/TIME COST

### M.1   MEMORY ANALYSIS

Memory cost for Eval mode: $\mathcal{O}(X + Y) = \mathcal{O}(NC_{\text{in}}H_{\text{in}}W_{\text{in}} + NC_{\text{out}}H_{\text{out}}W_{\text{out}})$.

Memory cost for Tune mode: $\mathcal{O}(X + \omega') = \mathcal{O}(NC_{\text{in}}H_{\text{in}}W_{\text{in}} + k^2C_{\text{in}}C_{\text{out}})$.

For each ConvBN block, *the memory cost reduction of Tune mode* is $\mathcal{O}(NC_{\text{out}}H_{\text{out}}W_{\text{out}} - k^2C_{\text{in}}C_{\text{out}})$.

From the analysis, we can conclude that networks with larger feature maps (larger $H_{\text{out}}W_{\text{out}}$, such as HRNet that features high resolutions), smaller kernel sizes (smaller $k$, such as RepVGG with many $k = 1$ conv kernels), and larger batch sizes (larger $N$) will benefit more from the proposed Tune mode. The conclusion can be empirically validated from Table 8. We can observe that:

- The memory cost reduction ratio grows from $18.47\%$ to $34.83\%$ when batch size grows from 2 to 16, for the same Faster RCNN detector with ResNet101 backbone.

- The memory cost reduction ratio grows from $18.47\%$ to $35.76\%$ when changing the network backbone from ResNet101 to HRNet while keeping the rest the same.
- The memory cost reduction ratio grows from $34.83\%$ to $43.04\%$ when changing the network backbone from ResNet101 to RepVGG while keeping the rest the same.

## M.2 TIME ANALYSIS

As pointed out by the FlashAttention paper (Dao et al., 2022), *the computation time of modern GPU hardware often scales with memory access* (the number of bytes the program reads from and writes to memory). The cost of memory access can be effectively estimated by summing all node sizes from computation graphs in Table 2. Therefore, we can analyze the time cost of Eval mode and Tune mode as follows:

Eval mode time cost:

$$\mathcal{O}(X+Y+\omega+b+\bar{Y}+\hat{\mu}+\hat{\sigma}+\beta+\gamma+Z) = \mathcal{O}(NC_{\text{in}}H_{\text{in}}W_{\text{in}}+3NC_{\text{out}}H_{\text{out}}W_{\text{out}}+k^2C_{\text{in}}C_{\text{out}}+5C_{\text{out}}).$$

Tune mode time cost:

$$\mathcal{O}(X+\omega+b+\omega'+b'+\hat{\mu}+\hat{\sigma}+\beta+\gamma+Z) = \mathcal{O}(NC_{\text{in}}H_{\text{in}}W_{\text{in}}+NC_{\text{out}}H_{\text{out}}W_{\text{out}}+2k^2C_{\text{in}}C_{\text{out}}+6C_{\text{out}}).$$

For each ConvBN block, *the time cost reduction of Tune mode* is:

$$\mathcal{O}(2NC_{\text{out}}H_{\text{out}}W_{\text{out}} - k^2C_{\text{in}}C_{\text{out}} - C_{\text{out}}).$$

Since feature maps (size $NC_{\text{out}}H_{\text{out}}W_{\text{out}}$) are typically much larger than convolutional kernels (size $k^2C_{\text{in}}C_{\text{out}}$), the proposed Tune mode can reduce time cost.

## N INTEGRATION WITH DL COMPILERS AND COMMON LIBRARIES

Our algorithm has been integrated into PyTorch core, MMCV, and MMEngine. We also support standalone usage.

### N.1 PYTORCH

Our method has been integrated into PyTorch core since version 2.2. People using PyTorch can turn on the Tune mode via the following code:

```
torch._inductor.config.efficient_conv_bn_eval_fx_passes = True
```

### N.2 MMCV/MMENGINE

Our method has been integrated into popular computer vision libraries. MMCV holds core operators, while MMEngine is the training framework. People using MMCV/MMEngine can turn on the Tune mode via adding a command line argument:

```
--cfg-options efficient_conv_bn_eval="[backbone]"
```

### N.3 STANDALONE USAGE

For people using old versions of PyTorch (we require PyTorch larger than 1.8), they can turn on Tune mode via the online code.

```
model = MyModel() # init a model
import tune_mode_convbn
tune_mode_convbn.turn_on_efficient_conv_bn_eval_for_single_model(model)
# now this model can benefit from tune mode, if it is trained with `Eval` mode.
```

