# OpenReview forum: "Efficient ConvBN Blocks for Transfer Learning and Beyond"
_ICLR.cc/2024/Conference — ICLR 2024 spotlight_

### Official Review · Reviewer_iJzH · 2023-10-20

**Soundness:** 3 good
**Presentation:** 4 excellent
**Contribution:** 4 excellent
**Rating:** 8
**Confidence:** 4

**Summary:**

This paper explored the trade-off between stability and efficiency in Convolution-BatchNorm blocks which are popular convolution neural networks：Deploy mode is efficient but suffers from training instability; Eval mode is widely used in transfer learning but lacks efficiency.
Based on detailed analysis, the paper proposed a Tune mode, which is stable and efficient, to bridge the gap between Eval mode and Deploy mode. Numerous experiments conducted in various tasks have verified the effectiveness of the proposed Tune mode.

**Strengths:**

1. The paper provided detailed  analysis of proposed Tune mode from both theoretical and experimental perspectives.
2. The author's writing is very good, and the entire paper is relatively easy to understand.
3. Simple algorithm, easy to follow. (codes are available for the public according to the abstract: "Our method has been integrated into both PyTorch (general machine learning framework) and MMCV/MMEngine (computer vision framework).")
4. The experimental results are reliable and sufficient to verify the effectiveness of this method.

**Weaknesses:**

1. the proposed Tune mode looks like an engineering technique. (this method might be more suitable for patent applications)
2. The novelty of this method is somewhat weak for top-tie conferences focused on theoretical research.

**Questions:**

see weaknesses above

---

> ### Author Response · Authors · 2023-11-14
> **Response to Reviewer iJzH**
>
> Thank you for your appreciation. Here are our responses to your concerns:
>
> # This method might be more suitable for patent applications
>
> Indeed, we can file patent applications for the method. However, we decide to contribute the method directly to the community, so it can benefit the community without legal constraints.
>
> # Compatibility of this method in a top-tie conference focused on theoretical research like ICLR
>
> We believe the charm of ICLR as a top-tie conference is that it is open and inclusive, including theoretical and practical papers. Also note that our paper consists of not only a practical method (Tune Mode ConvBN), but also non-trivial analyses of the instability of Deploy mode and the equivalence between Tune mode and Eval mode in terms of forward-backward computation. Therefore, in our humble opinion, our paper should be a good fit for ICLR.
>
> We hope the above responses can address your concerns. Should you have any other questions, we will be happy to provide more information.

---

> > ### Comment · Reviewer_iJzH · 2023-11-21
> >
> > Thank you very much for your feedback and clarification. I agree with the technological contribution and practical significance.
> > Considering the author's response and additional contributions during the rebuttal, I have raised my rating.

---

> > > ### Author Response · Authors · 2023-11-21
> > > **Response to Reviewer iJzH**
> > >
> > > Thank you for your positive feedback and kind response that greatly encouraged us. Our team is actively working on integrating our method into widely used open-source libraries, ensuring it becomes more accessible and beneficial to the broader community. We are committed to continuous development and welcome any additional suggestions or feedback you may have.

---

### Official Review · Reviewer_1pnn · 2023-10-30

**Soundness:** 3 good
**Presentation:** 4 excellent
**Contribution:** 3 good
**Rating:** 6
**Confidence:** 4

**Summary:**

In this paper, the authors introduce the Tune mode of ConvBN blocks, which can save memory and time costs during transfer learning. Experiments show that when transferring the backbone with ConvBN to downstream tasks, Tune mode can save 20%-40% of memory costs without loss of accuracy.

**Strengths:**

1. The writing and presentation of the paper are clear and easy to understand.
2. The method (Tune mode for ConvBN blocks) is reasonable and novel.
3. The experiments on five datasets and 12 model architectures show the method's effectiveness: Tune mode can save 20%-40% of memory costs without losing accuracy.

**Weaknesses:**

1. Although the method to reduce the cost of Eval mode is very clever, the contribution of the paper to academic research is limited. This paper is more like introducing a useful technical trick to reduce the overhead of ConvBN's eval mode in transfer learning. There are no foreseeable follow-up research directions here. This method technically belongs to Gradient Checkpointing (Bulo et al. (2018)), but the authors cleverly use the affine transformation of BN to escape the time cost. On the other hand, this trick can only apply to BN with eval mode.

**Questions:**

1. The conclusion that "Eval mode gets significantly better mAP than Train mode" is too strong for me. From my personal experience, I sometimes get better accuracy using BN's training mode than using eval mode for transfer learning. Therefore, the authors should provide more experiments than the two selected models to verify this strong conclusion.
2. As SyncBN's eval mode has similar behavior to BN, I am not sure whether the Tune mode can be directly applied to the backbone trained with SyncBN. The author can explain this in the response.

---

> ### Author Response · Authors · 2023-11-14
> **Response to Reviewer 1pnn**
>
> Thank you for your appreciation. Here are our responses to your concerns:
>
> # Possible follow-up research directions
>
> There are two possible kinds of follow-up research:
>
> - To reduce further the memory/time cost on top of our Tune mode ConvBN blocks.
> - To apply our Tune mode ConvBN blocks to various domains.
>
> While we admit the first kind of follow-up research might be difficult, we firmly believe that **there will be a lot of follow-up research in the second direction**, especially in the computer vision and object detection community. For example, as shown in Figure 1 in the paper, our method can benefit 496 (78.2%) training configurations in the popular MMDetection library.
>
> In addition, as mentioned in [the common response](https://openreview.net/forum?id=lHZm9vNm5H&noteId=cgxGu9HEox), to the best of our knowledge, our paper is among the first batch of papers to bridge pioneering deep learning compilers with machine learning algorithms. In this aspect, our work can also inspire future research in the intersection of machine learning algorithms and systems.
>
> # The claim "Eval mode gets significantly better mAP than Train mode" is too strong
>
> Thanks for pointing it out. We will turn down the tone. The claim is stated in the context of object detection, where the batch size is often small and [Train mode is known to deteriorate](https://proceedings.neurips.cc/paper/2017/file/c54e7837e0cd0ced286cb5995327d1ab-Paper.pdf). Please refer to "Batch renormalization: Towards reducing minibatch dependence in batch-normalized models" in NeurIPS 2017 for details.
>
> # Whether the Tune mode can be directly applied to the backbone trained with SyncBN
>
> That's a nice catch! SyncBN's Eval mode has the same behavior as BN's Eval mode, so our method can be directly applied to the backbone trained with SyncBN.
>
> We hope the above responses can address your concerns. Should you have any other questions, we will be happy to provide more information.

---

> > ### Author Response · Authors · 2023-11-22
> > **We are looking forward to your feedback.**
> >
> > Dear reviewer,
> >
> > Following your questions, we clarified the follow-up research directions and the SyncBN applicability, as well as adjust the tone of a claim. We also highlight the [integration with deep learning compiler](https://openreview.net/forum?id=lHZm9vNm5H&noteId=cgxGu9HEox) and [the impact on open-source community](https://openreview.net/forum?id=lHZm9vNm5H&noteId=KwTYteJUnC) in the common response notes, which further confirms that there would be many follow-up application research works using our method.
> >
> > We eagerly wait for your valuable feedback on the above response.

---

### Official Review · Reviewer_nwrB · 2023-11-01

**Soundness:** 4 excellent
**Presentation:** 4 excellent
**Contribution:** 3 good
**Rating:** 8
**Confidence:** 4

**Summary:**

The paper proposes a novel calculation strategy for ConvBN by changing the order of calculation. The proposed Tune mode have exact same forward/back propagation expression with the eval mode yet are faster and has lesser memory footprint, demonstrated both theoretically and empirically. This makes it a useful component in transfer learning. The authors also attributed the instability of the training using deploy mode to the scaled weight and gradient, leading to the necessity of saving part of the parameters.

**Strengths:**

1. The writing is pretty clear with solid proof and extensive result.
2. The proposed method can be used widely in computer vision tasks.

**Weaknesses:**

Not much.

**Questions:**

Is it possible to merge $b$ and $\beta$?

---

> ### Author Response · Authors · 2023-11-14
> **Response to Reviewer nwrB**
>
> Thank you for your appreciation. Here is our response to your question:
>
> # Is it possible to merge $b$ and $\beta$?
>
> Yes, it is possible to merge $b$ and $\beta$. When people use ConvBN blocks, they usually merge $b$ and $\beta$. Examples include [torchvision](https://github.com/pytorch/vision/blob/main/torchvision/models/resnet.py#L49) and [timm](https://github.com/huggingface/pytorch-image-models/blob/main/timm/models/resnet.py#L92), two widely used library for computer vision that sets `bias=False` in ConvBN blocks.
>
> Our analysis targets the general case of unmerged $b$ and $\beta$. It also works for merged $b$ and $\beta$ (by setting $b=0$).
>
> We hope the above response can address your concern. Should you have any other questions, we will be happy to provide more information.

---

> > ### Comment · Reviewer_nwrB · 2023-11-21
> > **Response to rebuttal**
> >
> > Thanks for the response. I will keep my score.

---

> > > ### Author Response · Authors · 2023-11-21
> > > **Response to Reviewer nwrB**
> > >
> > > Thank you for your positive feedback and kind response that greatly encouraged us. Our team is actively working on integrating our method into widely used open-source libraries, ensuring it becomes more accessible and beneficial to the broader community. We are committed to continuous development and welcome any additional suggestions or feedback you may have.

---

### Official Review · Reviewer_E7re · 2023-11-09

**Soundness:** 3 good
**Presentation:** 3 good
**Contribution:** 3 good
**Rating:** 8
**Confidence:** 3

**Summary:**

This paper primarily investigates the forward computation and backpropagation process in model training or testing of three modes (i.e., Train, Eval, Deploy) of the ConvBN module (Convolution layer plus BN layer), with a focus on the instability of the Deploy mode during transfer learning. Thus, a new mode called Tune mode is proposed, which boasts the advantages of training stability and low computational time and space costs.

To verify the effectiveness of Tune mode, the authors selected 5 datasets and 12 models, and conducted extensive experiments in classification and detection tasks. The experiments show that using Tune mode for transfer learning results in a slight improvement in model performance, and a significant reduction in time and space costs. Furthermore, the authors tested the effectiveness of Tune mode in the generation of adversarial samples, thereby demonstrating the general applicability of Tune mode as a replacement for Eval mode in tasks that originally used Eval mode.

**Strengths:**

1. The article is clearly articulated, with detailed experimental settings, and the proposed Tune mode is simple to implement and easy to reproduce.
2. The theoretical analysis and experimental verification of the instability in Deploy mode training presented in the article seem plausible.
3. The author conducted a large number of experiments to validate the advantages of Tune mode in terms of training stability and low time-space cost, with the latter being significantly beneficial.

**Weaknesses:**

1. The method proposed in this article is only applicable to methods that originally used Eval mode, its application scenarios are not very broad, and the impact is relatively small.
2. This article only provides experimental verification for the time and space cost advantages of Tune mode. Is it possible to make theoretical estimates and give a result similar to O(N)?

**Questions:**

See above.

---

> ### Author Response · Authors · 2023-11-14
> **Response to Reviewer E7re**
>
> Thank you for your appreciation. Here are our responses to your concerns:
>
> # Scope and impact
>
> Our work focuses on basic building blocks (ConvBN blocks) in computer vision. As shown in Figure 1 of the paper, training ConvBN blocks with Eval mode takes up a substantial proportion (78.2%) of object detection models, underscoring our work's widespread applicability and impact especially in computer vision and object detection.
>
> # Theoretical analyses of benefit in memory/time cost
>
> ## Memory analysis
>
> The memory analysis is provided in Section 3.4.2 in the paper, and we rephrase it using the big-O notation required by reviewer E7re:
>
> Memory cost for Eval mode: $\mathcal{O}(X + Y) = \mathcal{O}(N C_{\text{in}} H_{\text{in}} W_{\text{in}} + N C_{\text{out}} H_{\text{out}} W_{\text{out}})$.
>
> Memory cost for Tune mode: $\mathcal{O}(X + \omega^\prime) = \mathcal{O}(N C_{\text{in}} H_{\text{in}} W_{\text{in}} + k^2C_{\text{in}} C_{\text{out}})$.
>
> For each ConvBN block, **the memory cost reduction of Tune mode** is $\mathcal{O}(N C_{\text{out}} H_{\text{out}} W_{\text{out}} - k^2C_{\text{in}} C_{\text{out}})$.
>
> From the analysis, we can conclude that networks with larger feature maps (larger $H_{\text{out}}W_{\text{out}}$, such as HRNet that features high resolutions), smaller kernel sizes (smaller $k$, such as RepVGG with many $k=1$ conv kernels), and larger batch sizes (larger $N$) will benefit more from the proposed Tune mode. The conclusion can be empirically validated. We take correponding results from Table 8 in the paper to present the following table for your convenience.
>
> We can observe that:
>
> - The memory cost reduction ratio grows from $18.47$% to $34.83 $% when the batch size grows from $2$ to $16$, for the same Faster RCNN detector with ResNet101 backbone.
> - The memory cost reduction ratio grows from $18.47$% to $35.76$% when changing the network backbone from ResNet101 to HRNet while keeping the rest the same (i.e., mainly enlarge $H_{\text{out}}W_{\text{out}}$ ).
> - The memory cost reduction ratio grows from $34.83$% to $43.04$% when changing the network backbone from ResNet101 to RepVGG while keeping the rest the same (i.e., mainly decrease $k$).
>
> | Detector | Backbone | Batchsize | Precision | mode | mAP | Memory(GB)|
> |---------|---------|---------|---------|---------|---------|---------|
> | Faster RCNN  | ResNet101  | 2  | FP16  | Eval  | 0.3944  | 3.849   |
> | Faster RCNN  | ResNet101  | 2  | FP16  | Tune  | 0.3925  |  3.138 ($\textbf{18.47}$%$\downarrow$)   |
> | Faster RCNN  | ResNet101  | 8  | FP16  | Eval  | 0.3922  | 10.411   |
> | Faster RCNN  | ResNet101  | 8  | FP16  | Tune  | 0.3917  |  7.036  ($\textbf{32.41}$%$\downarrow$)  |
> | Faster RCNN  | ResNet101  | 16  | FP16  | Eval  | 0.3902  | 19.799   |
> | Faster RCNN  | ResNet101  | 16  | FP16  | Tune  | 0.3899  |  12.901  ($\textbf{34.83}$%$\downarrow$)  |
> | Faster RCNN  | HRNet  | 2  | FP32  | Eval  |  0.4017 | 8.504   |
> | Faster RCNN  | HRNet  | 2  | FP32  | Tune  |  0.4031 | 5.463  ($\textbf{35.76}$%$\downarrow$)   |
> | Faster RCNN  | RepVGG  | 16  | FP16  | Eval  |  0.3350 | 15.80   |
> | Faster RCNN  | RepVGG  | 16  | FP16  | Tune  |  0.3350 | 9.00 ($\textbf{43.04}$%$\downarrow$)  |
>
> ## Time analysis
>
> As pointed out by the [FlashAttention paper](https://proceedings.neurips.cc/paper_files/paper/2022/hash/67d57c32e20fd0a7a302cb81d36e40d5-Abstract-Conference.html) ("Flashattention: Fast and memory-efficient exact attention with io-awareness", in NeurIPS 2022), **the computation time of modern GPU hardware often scales with memory access** (the number of bytes the program reads from and writes to memory). The cost of memory access can be effectively estimated by summing all node sizes from computation graphs in Table 2. Therefore, we can analyze the time cost of Eval mode and Tune mode as follows:
>
> Eval mode time cost: $\mathcal{O}(X + Y + \omega + b + \bar{Y} + \hat{\mu} + \hat{\sigma} + \beta + \gamma + Z) = \mathcal{O}(N C_{\text{in}} H_{\text{in}} W_{\text{in}} + 3 N C_{\text{out}} H_{\text{out}} W_{\text{out}} + k^2C_{\text{in}} C_{\text{out}} + 5 C_{\text{out}})$.
>
> Tune mode time cost: $\mathcal{O}(X + \omega + b + \omega^\prime + b^\prime + \hat{\mu} + \hat{\sigma} + \beta + \gamma + Z) = \mathcal{O}(N C_{\text{in}} H_{\text{in}} W_{\text{in}} + N C_{\text{out}} H_{\text{out}} W_{\text{out}} + 2 k^2C_{\text{in}} C_{\text{out}} + 6 C_{\text{out}})$.
>
> For each ConvBN block, **the time cost reduction of Tune mode** is $\mathcal{O}(2 N C_{\text{out}} H_{\text{out}} W_{\text{out}} - k^2C_{\text{in}} C_{\text{out}} - C_{\text{out}})$. Since feature maps (size $N C_{\text{out}} H_{\text{out}} W_{\text{out}}$) are typically much larger than convolutional kernels (size $k^2C_{\text{in}} C_{\text{out}}$) and biases (size $C_{\text{out}}$), the proposed Tune mode can reduce time cost.
>
> We hope the above responses address your concerns. Should you have any other questions, we will be happy to provide more information.

---

> ### Author Response · Authors · 2023-11-22
> **We are looking forward to your feedback.**
>
> Dear reviewer,
>
> Following your questions, we added detailed theoretical analyses for the memory benefit and time cost reduction. We also highlight the [integration with deep learning compiler](https://openreview.net/forum?id=lHZm9vNm5H&noteId=cgxGu9HEox) and [the impact on open-source community](https://openreview.net/forum?id=lHZm9vNm5H&noteId=KwTYteJUnC)  in the common response notes.
>
> We eagerly wait for your valuable feedback on the above response.

---

### Author Response · Authors · 2023-11-14
**We would like to highlight a contribution that reviewers overlooked**

We thank all reviewers for appreciating our work. We want to highlight one contribution (the last contribution in the Introduction section) that seems to be overlooked:

> Our method has been quickly integrated into open-source framework libraries like PyTorch and MMCV/MMEngine because of its evident benefit, improving the efficiency of hundreds of models for everyone using these frameworks.

For those who are using a fresh new version of PyTorch (e.g., execute `conda install pytorch torchvision torchaudio pytorch-cuda=12.1 -c pytorch-nightly -c nvidia` to get the latest PyTorch version), they will find **our method is available in the PyTorch core library**, just like those built-in modules [nn.LayerNorm](https://pytorch.org/docs/stable/generated/torch.nn.LayerNorm.html) and [nn.BatchNorm2d](https://pytorch.org/docs/stable/generated/torch.nn.BatchNorm2d.html) people use every day. Note that PyTorch acknowledges our paper in their code, but we can't include the link to the implementation due to the conference policy of double-blind review requirements.

By integrating with recent advances in deep learning compilers, our method can benefit users **without changing their code**. Take the following code, for example:

```python
import torch
import torch.nn as nn

# define model architecture
class WrappedBatchNorm(nn.Module):
    def __init__(self):
        super().__init__()
        self.mod = nn.BatchNorm2d(128)
    def forward(self, x):
        return self.mod(x)

class M(nn.Module):
    def __init__(self):
        super().__init__()
        self.conv1 = nn.Conv2d(128, 128, 1)
        self.bn1 = nn.BatchNorm2d(128)
        self.conv2 = nn.Conv2d(128, 128, 1)
        self.nested = nn.Sequential(
            nn.BatchNorm2d(128),
            nn.Conv2d(128, 128, 1),
        )
        self.wrapped = WrappedBatchNorm()

    def forward(self, x):
        # self.conv1 + self.bn1 consist of one ConvBN block
        x = self.conv1(x)
        x = self.bn1(x)
        # self.conv2 + self.nested.0 consist of one ConvBN block
        x = self.conv2(x)
        x = self.nested(x)
        # self.nested.1 + self.wrapped.mod consist of one ConvBN block
        x = self.wrapped(x)
        return x

model = M()

model.eval().cuda()

input = torch.randn(64, 128, 224, 224).cuda()

# measure the benefit of enabling Tune Mode ConvBN Block

from torch._inductor import config as inductor_config
# enable the configuration to use our method. The proposed "Tune Mode ConvBN Block" is named "efficient conv bn eval" in PyTorch.
for enabled in (True, False):
    print(f"Tune Mode ConvBN Block enabled: {enabled}")
    torch._dynamo.reset()
    torch.cuda.reset_peak_memory_stats()
    inductor_config.efficient_conv_bn_eval_fx_passes = enabled
    compiled_model = torch.compile(model)

    # warm up cuda devices by computing both forward and backward
    compiled_model(input).sum().backward()

    # measure the time and memory cost
    import time
    start_time = time.time()
    start_memory = torch.cuda.memory_allocated()
    for i in range(10):
        compiled_model(input).sum().backward()
    time_cost = time.time() - start_time
    memory_cost = torch.cuda.max_memory_allocated() - start_memory
    print(f"time cost: {time_cost} seconds")
    print(f"memory cost: {memory_cost / 1024 / 1024} MB")
```

The code produces the following output (using RTX 4090 GPU), which shows the benefit of our method (58% less time cost and 25% less memory cost):

```
Tune Mode ConvBN Block enabled: True
time cost: 1.0495944023132324 seconds
memory cost: 9432.76513671875 MB
Tune Mode ConvBN Block enabled: False
time cost: 2.4892385005950928 seconds
memory cost: 12569.3427734375 MB
```

The only change to use our method is a configuration flag `inductor_config.efficient_conv_bn_eval_fx_passes = True`.

Moreover, the above code shows that **our method can directly benefit anyone using PyTorch without changing their code, even if the ConvBN patterns are very complicated** (there are three ConvBN blocks in the above code: `self.conv1` + `self.bn1`; `self.conv2` + `self.nested.0`; `self.nested.1` + `self.wrapped.mod`. The latter two pairs are non-trivial to discover and optimize).

To the best of our knowledge, our paper is among the first batch of papers to **bridge pioneering deep learning compilers with machine learning algorithms**, so that the whole PyTorch community can easily enjoy the efficiency gain brought by our method.

We hope the above clarification is clear to reviewers. Should you have any other questions, we will be happy to provide more information.

---

### Author Response · Authors · 2023-11-20
**We would like to share one more use case of our method from the community.**

Dear Reviewers,

We are writing to provide an update on the practical applications of our method.

Recently, we have received positive feedback from the open-source community regarding our method's application in **rotated object detection**. Notably, it has helped reduce the memory footprint from 17.12GB to 11.57GB, achieving a **32% memory saving without affecting the model's performance**. Furthermore, our method is compatible with over 80% (90 out of 111) configurations in the rotated object detection library [MMRotate](https://github.com/open-mmlab/mmrotate). This efficiency gain requires only a minimal change, specifically adding the command line option `--cfg-options efficient_conv_bn_eval="[backbone]"`, as detailed in our supplementary material.

These new use cases reinforce the contribution we outlined in [our previous comment](https://openreview.net/forum?id=lHZm9vNm5H&noteId=cgxGu9HEox), demonstrating how our method can seamlessly integrate with users' existing code through deep learning compilers, without the need for altering network structure definitions.

We are eager to hear your thoughts and feedback on these responses. Should you have any questions or require further information, please do not hesitate to contact us. Your insights are invaluable to the advancement of our work.

Thank you for your time and consideration.

Sincerely,

Authors of submission 3142

---

### Meta-Review · Area_Chair_63dq · 2023-12-05

**Metareview:**

Four experts reviewed the paper, and all were positive about it. The rebuttal addressed the reviewers' questions and highlighted that they had incorporated the method into pytorch.

**Justification For Why Not Higher Score:**

The work's scope is small, though the execution is solid.

**Justification For Why Not Lower Score:**

All reviewers were positive about it.

---

### Decision · Program_Chairs · 2024-01-16

Accept (spotlight)